# From theory to practice: An integrated TTF-UTAUT study on electric vehicle adoption behavior

**Ayed Alwadain**[1], **Suliman Mohamed Fati**[2], **Kashif Ali**[3]*, **Rao Faizan Ali**[4]

**1** Computer Science Department, Community College, King Saud University, Riyadh, Saudi Arabia, **2** Information Systems Department, College of Computer and Information Sciences, Prince Sultan University, Riyadh, Saudi Arabia, **3** Faculty of Management Science (FOMS), UCP Business School, University of Central Punjab, Lahore, Pakistan, **4** Department of Software Engineering, University of Management and Technology, Lahore, Pakistan

\* kashifali009@gmail.com

**Funding:** This research is supported by Researchers Supporting Project number (RSP2023R309), 434 King Saud University, Riyadh, Saudi Arabia, and the funder/author has

## Abstract

In Industry 4.0, the adoption of new technology has played a major role in the transportation sector, especially in the electric vehicles (EVs) domain. Nevertheless, consumer attitudes towards EVs have been difficult to gauge but researchers have tried to solve this puzzle. The prior literature indicates that individual attitudes and technology factors are vital to understanding users' adoption of EVs. Thus, the main aim is to meticulously investigate the unexplored realm of EV adoption within nations traditionally reliant on oil, exemplified by Saudia Arabia. By integrating the "task technology fit" (TTF) model and the "unified theory of acceptance and usage of technology" (UTAUT), this research develops and empirically validates the framework. A cross-section survey approach is adopted to collect 273 valid questionnaires from customers through convincing sampling. The empirical findings confirm that the integration of TTF and UTAUT positively promotes users' adoption of EVs. Surprisingly, the direct effect of TTF on behavioral intentions is insignificant, but UTAUT constructs play a significant role in establishing a significant relationship. Moreover, the UTAUT social influence factor has no impact on the EVs adoption. This groundbreaking research offers a comprehensive and holistic methodology for unravelling the complexities of EV adoption, achieved through the harmonious integration of two well-regarded theoretical frameworks. The nascent of this research lies in the skilful blending of technological and behavioral factors in the transportation sector.

## Introduction

Environmental issues, which affect all forms of life on Earth, such as climate change, carbon neutrality, and the reduction of carbon emissions, have emerged as some of the most significant challenges in the growth of economies [1]. These problems have grown in importance throughout the history of economic development, crossing national boundaries and influencing the welfare of all living things [2]. The use of fossil fuels by industry and the transportation

significant role in drafting, analysis, and revision of the manuscript.

**Competing interests:** The authors have declared that no competing interests exist.

sector is the main contributor to environmental problems [3, 4]. Upadhyay and Kamble [1] argued that manufacturing and energy are not only major contributors to environmental issues but instead transportation sector is also held responsible for this. Mpoi et al. [5] argued that the transportation industry is accountable for 24 percent of the world's gross carbon dioxide ($CO_2$) emissions from energy sources such as fossil fuels. Moreover, the IEA [6] report highlighted that more than 75% of transport $CO_2$ emissions come from road vehicles. Similar to this, Tiseo [7] in a Statista report, passenger vehicles stood out as the major source of $CO_2$ emissions in 2020, accounting for a sizable 41% of all transportation-related emissions worldwide. EVs stand out as favorable and workable solutions in a concerted effort to address and combat environmentally harmful practises [1, 8].

Asadi et al. [8], EVs are expected to serve a dual function in this situation, reducing adverse ecological impacts and protecting priceless non-renewable fuel resources during their entire use. Additionally, Singh et al. [3] emphasise that EVs have the potential to be effective replacements for achieving the Sustainable Development Goals (SDGs) of the United Nations, which are best reflected by affordable and clean energy. The 17 SDGs set out by the UN, which are expected to be accomplished by 2030, serve as both a road map for improving the world and a solid foundation for upcoming initiatives in business and research. The International Energy Agency (IEA) reported an astounding increase in electric vehicle (EV) sales in 2021 compared to previous years, culminating in a ground-breaking record high of 6.6 million EVs sold, highlighting the unquestionable rise towards an electrified and sustainable future. Furthermore, the research emphasised that in 2021, sales of electric vehicles will account for almost 10% of all automobile sales worldwide. The global market value of EVs is projected to grow over 11-fold from today's levels of sale [6]. The increase in EVs is primarily led by developed countries like Norway (86%), Iceland (73%), and Sweden (43%) [6]. By contrast, the sale of EVs is still lagging in emerging and developing economies. The adoption of EVs in the Middle East region is in the nascent stage [9]. UAE and Saudi Arabia are becoming the early adopters of EVs in the region. However, the contribution of GCC countries is not very convincing in comparison with the rest of the world [7]. Additionally, the COVID-19 pandemic and the Russia-Ukraine war have further disrupted the EV markets, especially in emerging and developing economies. Conclusively, the global rise of EV adoption reveals a notable gap, especially in Middle East region, due to geopolitical and pandemic disruptions, emphasizing the urgency for targeted interventions to facilitate a more inclusive and sustainable EV transition.

Saudi Arabia is mainly dependent on the sale and consumption of fossil fuel energy sources, which has put considerable pressure on the country to reduce its $CO_2$ emission. According to King Abdullah Petroleum Studies and Research Centre, the transportation sector's share of $CO_2$ emission was 21% in 2018. [10] Rahman et al., [11] argued that renewable energy is not used on a large scale in Saudi Arabia for road transportation. Consequently, the transport sector is associated with high $CO_2$ and greenhouse emissions. In the Middle East region, Saudi Arabia has the highest number of vehicle owners. This increasing number of vehicles will significantly increase the energy demand in road transportation [11].

Undeniably the 21st century is witnessing the role of individual behavior toward EVs [12]. According to Hardman [13], incentives have the potential to accelerate the adoption of EVs, but their importance cannot be applied uniformly due to the variety of consumer incentives across various locations. Maybury et al. [2] also emphasised that the move towards EVs is a technical transformation that includes changes to both technology and customer acceptance. As a result, completing this change quickly and effectively is a difficult challenge. In a nutshell, despite the pressing need for Saudi Arabia to reduce CO emissions, there exists a significant gap in the country transition to EV, exacerbated by a historical dependence on fossil fuels, limited use of renewable energy in road transportation, and the challenges of aligning consumer

incentives with the complex dynamics of technological transformation and consumer acceptance.

The adoption of electric vehicles (EVs) in earlier academic works was largely influenced by well-established paradigms of conventional technology adoption theories, such as the UTAUT and the "Technology Acceptance Model" (TAM), which were highlighted by Zhang et al. [4], the "Stimulus-Organism-Response" (SOR) model, which was investigated by Upadhyay and Kamble [1], and the Theory of Planned. Nevertheless, Upadhyay and Kamble [1] emphasise, that recent literature shows a change in thinking, with academics increasingly supporting different theories, views, and frameworks to clarify the dynamics of technology adoption within the EV sector. The extant studies solely focusing on end-user perceptions of technology may not be enough. Therefore, the individual and technology characteristics better explain the customer's intention to adopt EVs. Prior literature explains the adoption concept in various domains such as automotive [14], banking [15, 16], education [17], healthcare [18], manufacturing [19], and tourism [20]. Amidst evolving EV adoption literature, there is a growing recognition for diverse framework beyond traditional theories, necessitating deeper exploration of individual and technology dynamics. There is a paucity of empirical evidence for the adoption of EVs. To address the problem and research gaps, the following research objectives have been developed.

- To explore whether there is a substantial impact of TTF on consumer's intention to buy electric vehicles (EVs).

- To determine that the UTAUT model significantly affects the customer's intention to purchase EVs.

- To determine that combining the TTF and UTAUT frameworks results in a deeper understanding of customer intention towards EVs.

After achieving the above study objectives, this research has both contribution and implications. This study contributes by integrating TTF and UTAUT framework for a comprehensive view of EV adoption. It addresses the literature gap by providing insights into challenges and opportunities in Middle East EV adoption, especially in Saudi Arabia. The empirical research fills a theoretical gap, offering valuable evidence for future research and practical application in the EV domain. In term of implications, the research underscores the importance of aligning technology with user preferences, urging managers and decision-makers to recognize key determinants for creating a favourable environment for EV adoption within a comprehensive framework.

## Theoretical framework & hypotheses development

### Theoretical background

The prior literature has indicated that a rage of theories influence individual decision to use technology. The review study of Dash et al. [21] argued that firms failed to adopt new technology because of lack of planning for change, poor vision & adoption, and ignoring customer expectations. Börstler et al. [22] claimed that it is a well-known issue that adoption decision are rarely driven by evidence. Thus, the current research is based on Unified Theory of Acceptance and Use of Technology (UTAUT) and Task-Technology Fit.

**Unified Theory of Acceptance and Use of Technology (UTAUT).** The past studies affirmed the existence of various theories pursuing to explain why individuals adopt technology. These theories, however, were restricted in their ability to adequately explain technology adoption because they only covered specific features and frequently overlapped in their

explanations [23]. The past studies highlighted that the Technology Acceptance Model (TAM) is considered as the base of UTAUT theory by Venkatesh et al. [24]. Furthermore, UTAUT theory was developed by removing redundant or equivalent elements from the eight most prevalent technology adoption models [25]. As proposed by Venkatesh et al. [24] the intention and usage of technology are influenced by four main factors: performance expectancy, effort expectancy, facilitating condition and social influence. The constructs performance expectancy refers to the degree to which individual believe that utilizing the technology will empower them to achieve optimal improvement in job performance [23, 26]. Whereas effort expectancy pertains to the ease of effort associated with the utilization of technology [23, 27]. On the other hand, facilitating conditions involve the extent to which individuals are confident that the current organizational and technical infrastructure is in place to support the utilization of the technology [23, 27]. Finally, the belief that individual actions related to the use of technology will be accepted and embraced by affluent individuals or peers is referred to as social influence [23, 26]. The review study of Dash et al., [21] argued that facilitating conditions emphasizes user behaviour and remaining three constructs focuses on usage behaviour. In other words, a crucial aspects of UTAUT is that the decision of individual or organization is based on social or competitive factors to adopt new technology [21].

The UTAUT theory is widely used and applied in diverse context, such as mobile commerce [28], healthcare [29], higher education [30–33], fintech [34], environmental studies [35], and electrical vehicle [4, 36]. In the current study context, the study of Zhang et al. [4] examined the adoption factors of EVs in China through UTAUT theory. Jain et al. [36] investigated the elements affecting the adoption of EVs in India using the theoretical underpinnings of the UTAUT theory. The UTAUT theory was adopted by Chaveesuk et al. [37] to analyse customer acceptance trends for autonomous vehicles in Thailand. Finally, Abbasi et al. [38] investigated the motivations for Malaysian users' acceptance of EVs by employing the UTAUT theory.

**Task-technology fit model (TTF).** Task-technology fit framework (TTF) evaluates the compatibility of technological resources with the requirement of specific task [39]. Gebauer and Ginsburg, [40] argued that individual will only embrace technology if it seamlessly integrates into their routines activities and enhances overall performance. According to Cai et al. [27] TTF consist of two factors: task characteristics and technology characteristics. Task characteristics encompasses the challenges that potential adopters need to manage when employing technology in a specific context [27]. Conversely, technology characteristics encompasses the benefits offered by the technology [27]. Sediyaningsih et al. [23] argued that whenever technical features are compatible and properly tailored, it encourages to adopt new technology. On the contrary, if technological features are not compatible, the technology usage is discouraged [23]. Dayour et al. [26] argued that technology must be utilized and good fit with the task to support performance. In the same vein, Börstler et al. [22] contended that TTF is closely linked to cognitive fit, suggesting that better performance results from a strong alignment between task and the mental representation of problem solver.

Predominantly, TTF was designed with organizational context in mind, based on technology and employee-handled work activities [39]. However, in the recent years, TTF has expanded beyond the workplace to encompass a wider range of context and technologies like satisfaction and continuance intention of EV drivers (Cruz-Jesus et al., 2023) [41], online gamification [42], e-service quality [43], m-health [44], and blockchain food delivery [45].

**Integration of UTAUT and TTF.** This research integrates the UTAUT and TTF to explain the user adoption of EVs from both perspective including TTF and technology perceptions. According to Zhou et al. [46] argued that individual variances of UTAUT (45.7%) and TTF (43.3%) lower than the integrated model of UTAUT-TTF (57.5%). This highlight the

advantage of integrated model over individual models. Khashan et al. [25] argued that the integration of both theories is crucial, emphasizing that user preference to adopt new technologies goes beyond mere perceptions. It also hinges on how effectively the characteristics of those technologies align with the tasks they are intended to assist users in completing. Several studies have found a strong association among TTF and UTAUT constructs, which may facilitate adoption in automotive [14], banking [15], construction [27], education [47], healthcare [48], and public sector [49] as shown in Table 1. Nevertheless, there is a dearth of empirical evidence supporting the application of the integrated framework in the EVs domain. This points to a theoretical gap in previous research, which can be addressed by incorporating it into the current study to fill this void.

## Hypotheses development

**Task and Technology characteristics (TAC & TEC).**   According to Zhou et al., [46], the term "task characteristics" (TAC) refers to essential elements of user task needs [50]. On the other hand, Al-Rahmi et al. [17], "technology characteristics" (TEC) pertain to how technology can be used to obtain the best alignment. Zhou et al., [46] highlighted the crucial significance of both task and technology features in supporting the acceptance of cutting-edge technology in the context of TTF. According to Wang et al. [51], consumers are more likely to assume that adopting new technology can enhance their performance when they see a harmonic relationship between technology capabilities and tasks. The positive effect of these elements on TTF, which ultimately supports personal behavioural objectives, is highlighted by prior studies. For instance, Zhou et al., [46] hypothesize that TAC and TEC have a positive effect on TTF. The prior research has a strong foundation for the link between TAC, TEC, and TTF, as shown by publications by Afshan and Sharif [50], Wang et al. [51], and Wang et al. [52]. The ensuing hypotheses have been established as a result.

**H1:** TAC positively affects TTF.

**H2:** TAC positively affects TTF.

**Table 1. Integration of UTAUT and TTF.**

| Theoretical framework | Industry/sector type | Country | Sampling technique | Summary of findings | References |
|---|---|---|---|---|---|
| UTAUT and TTF | Automotive | South Korea | None | Both UTAUT-TTF can server an essential role in predicting consumer behaviour. | [14] |
| | Banking | Lebanon and China | Probability and Non-probability | The outcomes highlight that both UTAUT and TTF are essential theoretical framework to explain the mobile banking user adoption. | [15, 46] |
| | Construction | China | None | The constructs of TTF have significant association with UTAUT variables. | [27] |
| | Education | India, China, Malaysia | Probability and Non-probability | The integration of information system theories (UTAUT-TTF) promote students active learning and enable them to exchange information and knowledge more efficiently. | [17, 47, 52, 56, 72, 74] |
| | Healthcare | China, Turkey, Iran, South Korea, Pakistan | Probability and Non-probability | The findings indicated that UTAUT and TTF help to understand the user intention to adopt healthcare services. | [18, 48, 51, 53, 57] |
| | Retail | USA | Probability | Both frameworks (UTAUT and TTF) positively explain the consumers intention to adopt technology-driven products. | [58] |
| | Public sector | Pakistan, Jordan, Yemen | Probability and Non-probability | Both UTAUT and TTF are the stronger predictors of BI than either UTAUT or TTF alone. | [49, 77, 78] |
| | Services | Tiawan | None | Both theoretical frameworks are important to adopt new and innovative technologies. | [79] |

**Task technology fit (TTF) and behavioral intention (BI).** Empirical evidence, in the opinion of Gebauer and Ginsburg [40], supports the notion that people are reluctant to immediately accept and use new technology whenever it fails to meet their demands or enhance their performance. Individual impressions of the technology are a recurrent theme in many theories regarding technology adoption. TTF makes it clear that people's attitudes toward technology are influenced by other factors as well, in addition to their technological preferences. Instead, they are being influenced by other technological perspectives, people may find the technology useful and consent to its adoption largely because of TTF, as shown by Gu et al. [53]. The fundamental tenet of TTF holds that people won't adopt technology until it properly supports their daily activities and enhances their usefulness [54]. According to Al-Rahmi et al. [17], and Gu et al., [53], TTF has a major impact on behavioural intention (BI), which is proved by the body of literature. The ensuing hypothesis has been proposed.

**H3:** TTF positively affects user behavioral intention of EVs.

**H4:** TTF positively affects performance expectancy (PE).

**Effort expectancy (EE) & performance expectancy (PE).** The term "effort expectancy" refers to how easy a user perceives utilising technology to be [51]. Performance expectancy, on the other hand, is the extent to which the adoption of technology is anticipated to boost users' effectiveness in accomplishing specific tasks [51]. In the overall picture of EVs, effort expectancy indicates how consumers perceive the ease with which EVs may be used for transportation. Previous scholarly debates have shown a favourable association between EE, PE, and users' behavioural intentions. For instance, Zhou et al., [46] argued that the acceptance of mobile banking among users in China is positively correlated with effort expectancy, performance expectancy, and user adoption. According to Wang et al. [51], effort expectancy affects both performance expectancy and the propensity to utilise medical wearable technologies efficiently. In a similar vein, Paulo et al. [55] concluded that, in the context of implementing mobile augmented reality for tourism in Portugal, EE holds substantial connections with PE and BI. Finally, Wan et al. [56] confirmed that effort expectancy shows a significant association with both PE and BI in the uptake of online courses. In conclusion, users are more likely to expect enhanced performance and show a stronger propensity to adopt EVs when they perceive utilising them to be simple and easy. As a result, a hypothesis is put forward, relying on prior research.

**H5:** PE has a positive relationship with the intention of adopting EVs.

**H6:** EE positively affects performance expectancy.

**H7:** EE has a positive relationship with BI.

**Social influence (SI) and BI.** According to Wang et al. [51], social influence (SI) refers to a person's sense of how important others support their particular behaviours. According to Zhou et al. [46], this construct, which is comparable to subjective norms, reflects the influence of environmental circumstances. According to Wang et al. [51], the fundamental idea behind social influence is that people frequently attempt to strengthen their relationships with significant others by aligning their behaviours with these others' points of view. As evidenced in research studies by Alazab et al., [19], Afshan & Sharif [50], and Abdekhoda et al., [57], the body of existing literature repeatedly shows that social influence serves as a substantial predecessor to technology acceptance. If their behaviours are endorsed by social groups, users are more likely to embrace EVs. The ensuing hypothesis has been developed as a result.

**H8:** Social influence has a positive relationship with BI.

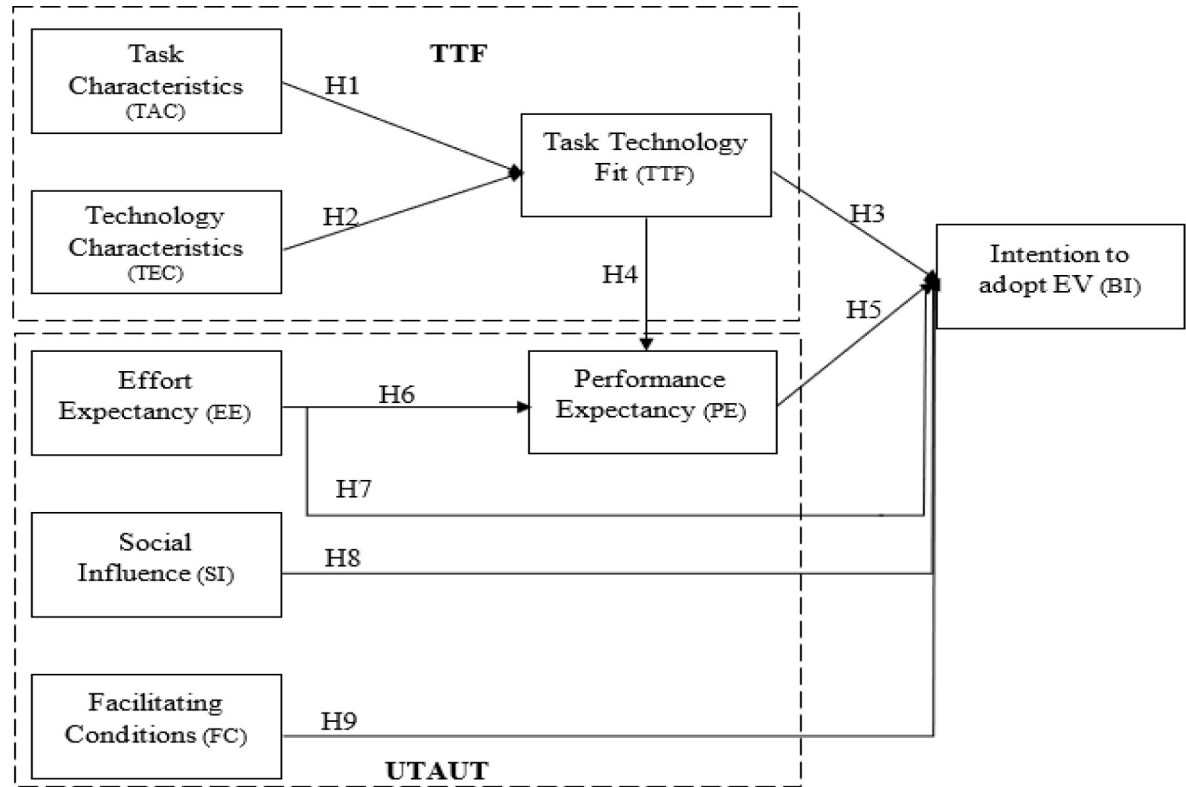

**Fig 1. Research model.**

**Facilitating conditions (FC) and BI.**   According to Wang et al., [51], facilitating conditions (FC) include a person's view of how much the organisational and technological infrastructure promotes the use of a system. By drawing a comparison, Zhou et al. [46] claimed that perceived behavioural control and FC are related concepts that capture the influence of a user's knowledge, resources, and skills. In the context of electric vehicles (EVs), FC refers to the expertise, knowledge, and support necessary to efficiently run EVs. Previous academic studies have convincingly shown that favourable circumstances are crucial in enhancing the behavioural intention to adopt new technology. For instance, studies by Wang et al., [51], Park et al., [58], and Zhao & Bacao [59] all indicated that favourable circumstances greatly increase people's propensity to embrace novel technologies. The ensuing theory has been stated as a result. Additionally, Fig 1 highlights the research model.

**H9:** Facilitating conditions positively affect BI to accept EVs.

## Research methodology

### Data collection and analysis

The selection of the sample population and method are based on study objectives. The data was collected from the central and western regions of Saudi Arabia. Elshurafa and Peerbocus [60] affirmed that more than 60% population lives in these regions. The online questionnaire was sent to the potential respondents through the convenience sampling method. Similar to the past studies [51], the inclusion criteria included the adults who owned EVs and the potential users. The current study delved into the acceptance of EVs from the viewpoints of both EV

**Table 2. Demographic statistics (N = 273).**

| Items | Sub-items | frequency | Percentage |
|---|---|---|---|
| Gender | Female | 147 | 53.85% |
| | Male | 126 | 46.15% |
| Age | 18–25 | 83 | 30.40% |
| | 26–35 | 97 | 35.53% |
| | 36–50 | 76 | 27.84% |
| | 51 or above | 17 | 6.23% |
| Education | Diploma | 89 | 32.60% |
| | Bachelors | 152 | 55.68% |
| | Postgraduate | 32 | 11.72% |
| Occupation | Students | 111 | 40.66% |
| | Employed | 105 | 38.46% |
| | Others | 57 | 20.88% |
| Region | Central | 138 | 50.55% |
| | Western | 135 | 49.45% |

users and non-users. 293 of the 708 delivered surveys were returned. A total of 273 valid questionnaires were considered suitable for data analysis after incomplete responses were excluded, yielding a response rate of 38.56%. Additionally, a pre-testing and pilot test were carried out before the questionnaires were distributed. Table 2 provides comprehensive demographic information for the entire group.

All participants received explicit information detailing the nature of their involvement, in accordance with ethical guidelines. Furthermore, the written informed consent from all the potential participants have been obtained for the inclusion in the study. This included questions regarding their personal information, demographics, and opinions on the use of electric cars (EVs). It was made clear to participants that they might answer the questionnaire anonymously. Additionally, their private information was handled with the strictest confidence both during and after the survey and was only used for research. As a result, our study adhered to ethical norms and handled any potential privacy-related issues without going beyond acceptable ethical bounds.

## Measurement

The eight variables that make up the study framework are all measured using a variety of items. According to Zhou et al., [46], the majority of these elements were extracted from earlier academic works to guarantee the conservation of content validity. Technology Characteristics (TEC) and Task Characteristics (TAC), each evaluated by four items, make up the components of Task-technology fit (TTF). These ideas were taken from Kang et al., [18] and Zhou et al., [46], respectively. PE, SI, FC, and EE are the four components of the UTAUT theory. These UTAUT variables' corresponding items were modified from Abdekhoda et al., (2022). Lastly, four items adapted from Zhang et al., [4] were used to examine behavioural intention. As indicated by Paulo et al. [55], a 7-point Likert scale was used, with response alternatives that ranged from "1" (strongly disagree) to "7" (strongly agree) [61].

## Analysis

### Assessing multivariate assumptions and CMB

Before performing the data analysis, multivariate assumptions were performed to confirm the dataset's quality. Hair et al., [62] recommended that in multivariate technique, the dataset

must satisfy the statistical assumptions like outlier, multicollinearity, and normality. To analyze the outlier, Mahalanobis D2 was used to detect multivariate outliers. The threshold value of D2 is higher than 4 [63]. In terms of multicollinearity, full collinearity analysis was performed as recommended by Hair et al. [64]. The study results highlight that all constructs' values are less than 5 [65]. Finally, the normality analysis was performed through skewness and kurtosis, all the values fall between ±2 [66].

Moreover, common method bias (CMB) can potentially affect research findings in behavioral sciences. As recommended by Podsakoff et al., [67], a Harman Single-factor test was performed to analyze CMB and if a single factor variance is less than 50% then CMB is not an issue. The results depicted that a single factor could explain the maximum variance was approximately 24%. Thus, CMB is not an issue in this study.

## Structural equation modeling (SEM)

In accordance with Hair et al., [65], the "partial least square structural equation modeling" (PLS-SEM) approach was adopted through SmartPLS. PLS-SEM provides consistent and comprehensive explanations of actual phenomena and this technique is excellent to other techniques like regression analysis [64, 65]. The analytical process of PLS-SEM consists of measurement and structural models.

**Measurement model.** The measurement model describes the constructs' reliability and validity statistics. In terms of validity, both convergent and discriminant validities were reported in the following sections. Additionally, the factor loading and variance inflated factor (VIF) results were reported.

*Reliability and convergent validity.* Table 3 exhibits the results for convergent validity and reliability for an array of constructs along with the corresponding items. Two indicators, composite reliability (CR) and rho_A were used in the reliability assessment process. The recommended criterion for CR and rho_A was set at 0.70, indicating satisfactory reliability, in accordance with the recommendations given by Hair et al. [64]. However, it's important to remember that CR values between 0.60 and 0.70 are also regarded as acceptable, as suggested by Hair et al. [65].

In the PLS-SEM literature, average variance extracted (AVE) and item loading were the primary indicators to measure convergent validity [62]. The threshold value of items loading $\geq$ 0.708 was considered excellent [65]. However, if item loading falls between 0.4 to 0.7 and the value of AVE is $\geq$0.5 then the construct should be retained [64, 65]. Moreover, the threshold value of AVE is $\geq$0.5. Based on the Table 3 results, all the constructs and their corresponding items have achieved the minimum threshold limits.

*Discriminant validity.* Cross-loading, Fornell-Larker, and the Heterotrait-Monotrait ratio of correlation (HTMT) are the three methods that are frequently used to evaluate discriminant validity in the body of extant literature. Notably, new findings from Henseler et al. [68] suggest that determining discriminant validity in the setting of PLS-SEM may not be best addressed by cross-loading and Fornell-Larker. As a result, the HTMT criterion was chosen as the preferred strategy in the present study [69, 70]. The stipulated criterion for HTMT values is below 0.85 for different structures and lower than 0.90 for similar constructs, consistent with the recommendations of Hair et al. [65]. The discriminant validity findings for each study variable are summarised in Table 4. It is noteworthy that the HTMT values for all research variables continue to be below the cutoff point of 0.85.

**Structural model.** In the path estimation approach, the bootstrapping technique with the 5000 resampling method and one-tailed were adopted to estimate the statistical significance between the study variables (p<0.05 & 0.10). Table 5 highlights the structural model statistics.

**Table 3. Reliability and validity statistics.**

| Variables | Items | Factor loading | VIF | Reliability | | AVE |
|---|---|---|---|---|---|---|
| | | | | CR | rho_A | |
| TAC | TAC1 | 0.760 | 1.683 | 0.836 | 0.731 | 0.562 |
| | TAC2 | 0.814 | 1.993 | | | |
| | TAC3 | 0.780 | 1.848 | | | |
| | TAC4 | 0.634 | 1.090 | | | |
| TEC | TEC1 | 0.714 | 1.666 | 0.844 | 0.764 | 0.576 |
| | TEC2 | 0.737 | 1.708 | | | |
| | TEC3 | 0.817 | 1.812 | | | |
| | TEC4 | 0.764 | 1.693 | | | |
| EE | EE1' | 0.838 | 2.053 | 0.875 | 0.809 | 0.639 |
| | EE2' | 0.843 | 2.057 | | | |
| | EE3' | 0.833 | 1.997 | | | |
| | EE4' | 0.668 | 1.245 | | | |
| SI | SI1 | 0.723 | 1.628 | 0.839 | 0.748 | 0.567 |
| | SI2 | 0.754 | 1.685 | | | |
| | SI3 | 0.742 | 1.728 | | | |
| | SI4 | 0.791 | 1.808 | | | |
| FC | FC1 | 0.685 | 1.223 | 0.864 | 0.786 | 0.615 |
| | FC2 | 0.813 | 1.851 | | | |
| | FC3 | 0.807 | 1.848 | | | |
| | FC4 | 0.823 | 1.870 | | | |
| TTF | TTF1 | 0.662 | 1.200 | 0.871 | 0.798 | 0.630 |
| | TTF2 | 0.833 | 2.067 | | | |
| | TTF3 | 0.820 | 1.943 | | | |
| | TTF4 | 0.846 | 2.194 | | | |
| PE | PE1 | 0.830 | 1.859 | 0.893 | 0.842 | 0.676 |
| | PE2 | 0.828 | 1.844 | | | |
| | PE3 | 0.805 | 1.793 | | | |
| | PE4 | 0.825 | 1.878 | | | |
| BI | BI1 | 0.865 | 1.030 | 0.708 | 0.856 | 0.593 |
| | BI2 | 0.681 | 1.669 | | | |
| | BI3 | 0.791 | 1.708 | | | |
| | BI4 | 0.786 | 1.721 | | | |

As expected, the association among task-technology-framework were positive and significant such as TAC→TTF (H1: β = 0.256, t-value = 6.198, p<0.001), and TEC→TTF (H2: β = 0.532, t-value = 13.812, p<0.001). Thus, H1 and H2 have been accepted. Contrary to expectations, the association of TTF→BI (H3: β = 0.086, t-value = 1.066, p = 0.146) was not statistically significant. Thus, H3 has not been supported. On the other hand, the association of TTF→PE (H4: β = 0.533, t-value = 13.275, p<0.001) was statistically and positively significant. Hence, H4 has been accepted. In term of UTAUT relationships, the association of EE→PE (H5: β = 0.328, t-value = 6.978, p<0.001) and EE→BI (H6: β = 0.150, t-value = 2.309, p<0.001) were statistically significant. Thus, H5 and H6 were supported. Unexpectedly, the association of SI→BI (H7: β = 0.079, t-value = 1.186, p = 0.121) was statistically insignificant. Hence, H7 has not been accepted. Furthermore, the relationship of FC→BI (H2: β = 0.113, t-value = 1.777, p = 0.041) was supported at a 90% confidence interval level. Thus, H8 has been accepted.

**Table 4. Discriminant validity.**

| Variables | TAC (1) | TEC(2) | EE (3) | SI (4) | FC (5) | TTF (6) | PE (7) | BI(8) |
|-----------|---------|--------|--------|--------|--------|---------|--------|-------|
| TAC (1) | | | | | | | | |
| TEC(2) | 0.726 | | | | | | | |
| EE (3) | 0.683 | 0.684 | | | | | | |
| SI (4) | 0.790 | 0.782 | 0.844 | | | | | |
| FC (5) | 0.709 | 0.745 | 0.675 | 0.659 | | | | |
| TTF (6) | 0.702 | 0.661 | 0.762 | 0.574 | 0.758 | | | |
| PE (7) | 0.799 | 0.794 | 0.792 | 0.761 | 0.574 | 0.791 | | |
| BI(8) | 0.430 | 0.494 | 0.459 | 0.538 | 0.470 | 0.476 | 0.558 | |

Finally, the association of PE→BI (H9: β = 0.336, t-value = 3.589, p<0.001) was positive and significant. Thus, H9 has been accepted.

Apart from the direct relationship, Table 5 indicates the indirect relationships. Based on the analysis, EE→PE→BI (β = 0.11, t-value = 3.273, p<0.001), TEC→TTF→PE→BI (β = 0.095, t-value = 3.448, p<0.001), and TAC→TTF→PE→BI (β = 0.046, t-value = 2.573, p<0.001) were positive and significant.

Moreover, Table 5 shows the effect size ($f2$). As recommended by Cohen (1988) the value of $f2$ above 0.02, 0.15, and 0.35 can be considered weak, moderate, and strong respectively (Hair et al., 2019). Based on the analysis, H1, H6, and H9 have a weak effect size, H5 has a moderate and H2 and H4 have a strong effect size.

Additionally, $R2$ facilitates the extent of the model's ability to explain a particular dependent variable. According to Cohen [71], $R2$ values of 0.02, 0.13, and 0.26, respectively, signify low, moderate, and large explanatory power. The results of the analysis show that TTF (0.504), PE (0.605), and BI (0.447) have significant coefficients of determination, which indicates large explanatory power as shown Table 6. Finally, Fig 2 highlights the structural model of this research.

## Discussion

The main goal of this study was to combine TTF and UTAUT frameworks to examine a theoretical model for the acceptance of EVs. When the integrated model was utilised, it explained

**Table 5. Path model analysis.**

| Relationships | β-value | t-value | p-value | f2 | Decision |
|---------------|---------|---------|---------|-----|----------|
| H1; TAC→TTF | 0.256 | 6.198 | 0.001 | 0.089 | Accepted |
| H2; TEC→TTF | 0.532 | 13.812 | 0.001 | 0.386 | Accepted |
| H3; TTF→BI | 0.086 | 1.066 | 0.146 | 0.005 | Not accepted |
| H4; TTF→PE | 0.533 | 13.275 | 0.001 | 0.448 | Accepted |
| H5; EE→PE | 0.328 | 6.978 | 0.001 | 0.170 | Accepted |
| H6; EE→BI | 0.150 | 2.309 | 0.013 | 0.020 | Accepted |
| H7; SI→BI | 0.079 | 1.186 | 0.121 | 0.004 | Not accepted |
| H8; FC→BI | 0.113* | 1.777 | 0.041 | 0.01 | Accepted |
| H9; PE→BI | 0.336 | 3.589 | 0.001 | 0.057 | Accepted |
| **Indirect effect** | | | | | |
| TAC → TTF → PE → BI | 0.046 | 2.573 | 0.007 | | Accepted |
| TEC → TTF → PE → BI | 0.095 | 3.448 | 0.001 | | Accepted |
| EE → PE → BI | 0.110 | 3.273 | 0.001 | | Accepted |

**Table 6. R2 and Q2 analysis.**

| Constructs | R2 | Q2 |
|---|---|---|
| TTF | 0.504 | 0.298 |
| PE | 0.605 | 0.479 |
| BI | 0.447 | 0.123 |

44.7% of the variation in BI to use EVs reported in Saudi Arabia. Furthermore, it was apparent from empirical examination that seven of the nine hypotheses proposed in this study were confirmed. The following explanation is offered for a more thorough understanding in line with the study's goals.

Assessing the value of the Task-technology fit (TTF) framework in revealing consumers' behavioural intentions to embrace EVs was the main goal of the initial study. Three hypotheses (H1–H3) were developed to accomplish this goal. The empirical results support the crucial roles that task and technological variables have in the TTF framework. These findings are in perfect conformity with earlier research from academic publications [47, 48, 51, 56, 72]. Liu et al. [73] contended that TTF is an essential predictors of BI. In the EV context, Cruz-Jesus et al. [41]) argued that TTF is an important predictor of intention then in internal combustion vehicles. The possible explanation of this is due to range of anxiety, which causes drivers to be

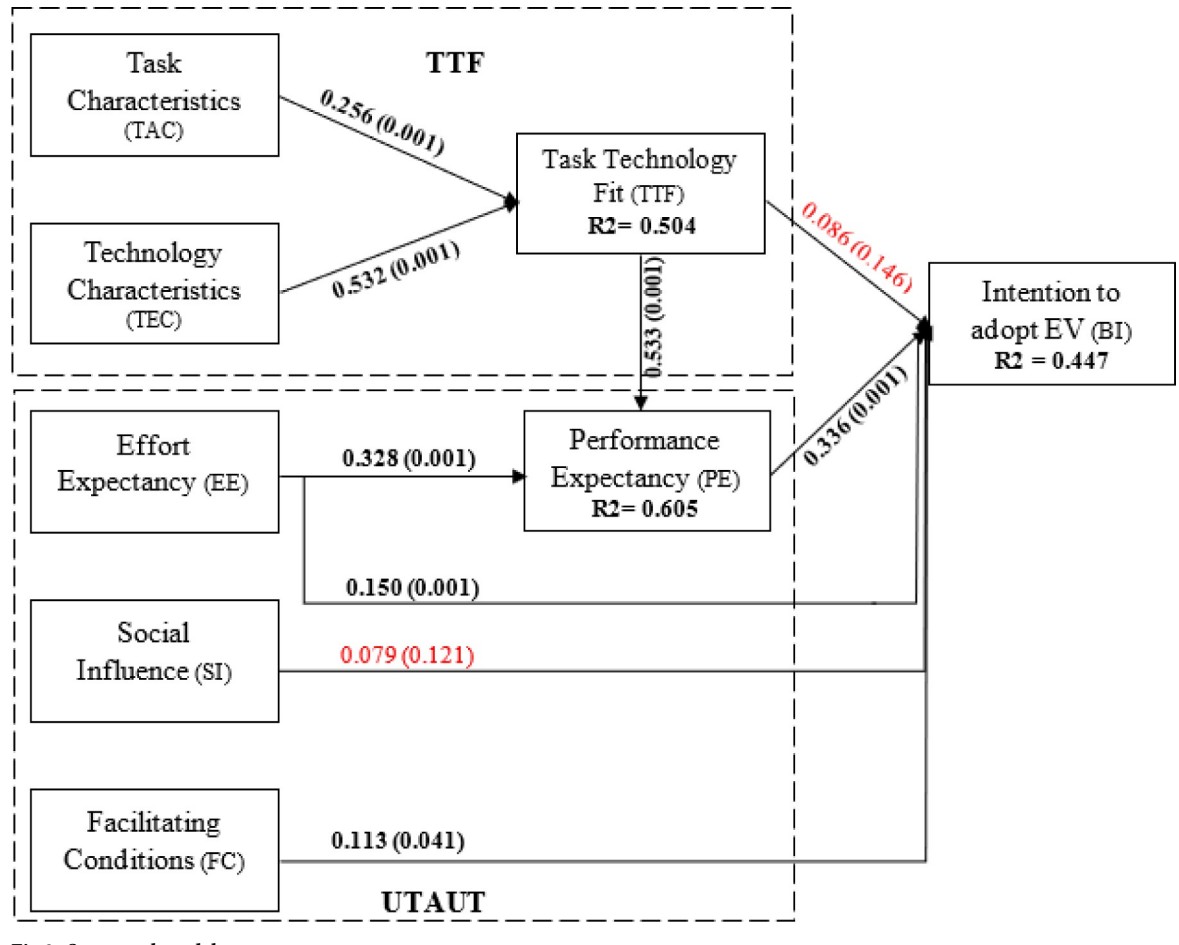

**Fig 2. Structural model.**

more cautious while using in-car amenities to conserve battery charge, which is intensified by the insufficiency of public infrastructure [41]. By taking into account EV manufacturers' strategic strategy, these results can be explained. Manufacturers must carefully evaluate the compatibility between consumers' work needs and the functions provided by EVs while marketing their EV models. For those who must travel frequently, as opposed to those who spend their time largely in offices, EVs may be a better option. While the latter could incline more towards conventional vehicles, the former group may find EVs to be a practical and appropriate option. Surprisingly, Wan et al. (2020) found no evidence to support the hypothesized relationship between TTF and behaviour. This might be explained by the fundamentally technological characteristics of EVs, which emerge in two essential dimensions. Customers can first customise vehicle features, such as style and colour, to suit their tastes. Second, there are many different functions that EVs can perform. $H_1$ and $H_2$ are supported by these arguments, while $H_3$ is rejected.

The UTAUT theory significantly increases user BI to embrace EVs, which is the second objective. To achieve this goal, five hypotheses (H5-H9) were developed. The empirical results show that, with the exception of social impact, all hypotheses between UTAUT factors and BI were marked by positive and statistically significant associations. The results support an earlier study [27, 53, 56, 57, 74] about the relationship between effort expectation (EE) and performance expectation (PE). Furthermore, the results are consistent with earlier research [25, 49, 59, 75] about the relationship between effort expectation (EE) and performance expectation (PE). Tian and Yang (2023) argued that performance expectancy has the most significant association with BI. The empirical findings show that Social Influence (SI) does not demonstrate a clear association with BI in the context of adopting EVs, in contrast to past studies. This result conflicts with the claims made by Wang et al. [51], but it is in line with the findings of Alazab et al. [19]. The adoption of existing EVs by consumers may not be strongly influenced by their consideration of other users' experiences, which is a reasonable explanation for this incongruity. Additionally, the participants might have more experience with and knowledge of technology, which would reduce their dependency on social networks. Last but not least, a favourable and statistically significant association between Facilitating Conditions (FC) and BI is found, a finding that is consistent with past academic research [50, 57].

The third objective emphasises that combining the TTF and UTAUT models provides a better understanding of how users would adopt electric vehicles (EVs). The influence of TTF on PE is where this integration is most visibly present. The empirical results show a significant and favourable connection between these constructs. These findings are consistent with previous research that has underlined the importance of technological and performance-related variables when it comes to integrating new technology [14, 23, 72, 73, 76]. Wan et al. [56] assert that an improved impression of the utility of EVs correlates with a higher degree of technological fit. Sediyaningsih et al., [23] argued that the integration of TTF and UTAUT theory have well-explained the BI to use metaverse. The combined use of TTF and UTAUT produces a thorough and effective model for identifying the elements that influence the adoption of new technology [27, 48, 57].

## Contributions and implications

**Theoretical contributions.** The research's findings offer significant new contributions and ramifications. First, this study emphasises the critical need to broaden our understanding of electric vehicles (EVs), which has primarily relied on a single theoretical framework, in response to a call in the literature [4, 38]. The TTF and UTAUT frameworks are combined in this research to offer a fresh viewpoint that enhances the debate on electric vehicles. Through a

methodical empirical approach, this work also aims to enhance the field of technological acceptance models. The validated model also reveals crucial associations with strong coefficients. Second, in addressing the identified literature gap, this research contributes by examining the specific challenges and opportunities in the Middle east region, especially in Saudia Arabia related to EV adoption. This study offers nuanced insights and proposes targeted interventions to foster a more inclusive and sustainable transition to EVs in this unique and dynamic environment. Finally, this empirical research makes a significant contribution by addressing the identified theoretical gap in EV adoption literature. By applying an integrated framework of UTAUT and TTF, the study offers empirical evidence to fill the existing void in research in the context of EV adoption. This contribution serves to enrich and validate the theoretical landscape, providing valuable insights for future research and practical applications in the EV domain.

**Practical implications.** Apart from theoretical contributions the research outcomes have threefold practical implications. First, at organization level the EV industry should prioritize initiatives that enhance TTF by optimizing in-car facilities, addressing rage of anxiety concerns, and improving the overall user experience. By doing so, they can positively influence consumer BI, and emphasizing the need for a strategic alignment between technology and user preferences. Second, manager and decision-makers should recognize the significance of UTAUT theory in shaping the customer BI to adopt EV. Strategies should be devised to align marketing and educational efforts with the key determinants identified in the UTAUT model, thereby fostering a favourable environment. Finally, acknowledging the value of combining TTF and UTAUT, organizations and policymakers should adopt an integrated approach in understanding customer intentions towards EVs. This involves the interplay between technical aspects of EV technology (TTF) and broader socio-psychological factors outlined in UTAUT. By implementing policies and strategies that considered both these dimensions can lead to a more holistic understanding of consumer behaviour, facilitating effective decision-making in the EV industry.

**Limitations and future directions.** While the study's limitations suggest captivating directions for further research, it does provide new insights into the world of transportation. Firstly, the TTF and (UTAUT) frameworks serve as the main pillars of the current study. Future research could include alternative theories like the perceived value theory and the success framework for information systems to expand the breadth of our knowledge. Second, a cross-sectional technique for data collection was used in this study. Because user behaviour displays dynamic and changing patterns, future studies could benefit from longitudinal study approaches. A more sophisticated comprehension of how users' adoption behaviours change over time would be provided by such a method. Thirdly, the sample size was impacted by restrictions brought on by the current study's resource limits. A suggested approach would involve repeating a similar framework with a larger and more varied sample size to increase the study's robustness. Fourthly, although moderating variables including context (e.g., age, gender), technology trust, and other aspects were not fully examined, the research conclusions are limited. Future research could explore a thorough approach that takes these moderating factors into account. The study's scope is also limited to Saudi Arabia, where EV adoption is still in its infancy. Future research might think about conducting a comparison study that compares early-stage and advanced-stage EV adoption to broaden the study's focus and provide cross-context insights.

**Conclusion.** In conclusion, the study successfully integrated the TTF and UTAUT framework to comprehensively examine the adoption of EV. The integrated model explained 44.7% of the variation to BI to adopt EV in Saudi Arabia, confirming seven out of nine hypotheses. The findings underscore the critical role of TTF, the significant impact of UTAUT factors on BI, and the enhanced understanding achieved through combining both models. The study

insights have practical implications for EV manufacturers and policy makers, emphasizing the need for strategic alignment with user preferences and a holistic approach to technology adoption.

## Supporting information

**S1 File. Past studies on TTF and UTAUT relationship.** This file highlights the past literature studies on TTF and UTAUT relationship.
(DOCX)

## Author Contributions

**Conceptualization:** Ayed Alwadain, Kashif Ali.

**Data curation:** Kashif Ali.

**Formal analysis:** Suliman Mohamed Fati, Kashif Ali.

**Funding acquisition:** Ayed Alwadain, Suliman Mohamed Fati.

**Investigation:** Kashif Ali.

**Methodology:** Ayed Alwadain, Suliman Mohamed Fati, Kashif Ali, Rao Faizan Ali.

**Project administration:** Ayed Alwadain, Suliman Mohamed Fati.

**Resources:** Suliman Mohamed Fati, Rao Faizan Ali.

**Software:** Ayed Alwadain, Rao Faizan Ali.

**Supervision:** Ayed Alwadain, Suliman Mohamed Fati.

**Validation:** Ayed Alwadain, Suliman Mohamed Fati, Rao Faizan Ali.

**Visualization:** Rao Faizan Ali.

**Writing – original draft:** Ayed Alwadain, Kashif Ali.

**Writing – review & editing:** Suliman Mohamed Fati, Rao Faizan Ali.

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
