## [Decision Letter · Decision Letter 0]

6 Nov 2023

PONE-D-23-27266From Theory to Practice: An integrated TTF-UTAUT study on Electric Vehicle Adoption BehaviorPLOS ONE

Dear Dr. Ali,

Thank you for submitting your manuscript to PLOS ONE. After careful consideration, we feel that it has merit but does not fully meet PLOS ONE’s publication criteria as it currently stands. Therefore, we invite you to submit a revised version of the manuscript that addresses the points raised during the review process.

You need to justify why the two models to be integrated for adoption of EV studies.

We look forward to receiving your revised manuscript.

Kind regards,

Sudarsan Jayasingh, Ph.D

Academic Editor

PLOS ONE

Journal Requirements:

"This research is supported by Researchers Supporting Project number (RSP2023R309), 434 King Saud University, Riyadh, Saudi Arabia"

"NO authors have competing interests"

Additional Editor Comments:

The research gap is not presented. What research gap exists related to study of adoption of EV. Objective need to be focused on identifying the factors related to adoption of EV. You have listed nine hypotheses but covered it in your objectives of the study. Model integration can be one of the objectives. Justification need to provide why you want to link for adoption of EV studies.

Discussion part did not explain the findings of your hypothesis. Need to improve the discussion part.

Contribution of research is not presented.

Reviewers' comments:

Reviewer's Responses to Questions

**Comments to the Author**

1. Is the manuscript technically sound, and do the data support the conclusions?

Reviewer #1: No

Reviewer #2: Yes

Reviewer #3: Yes

2. Has the statistical analysis been performed appropriately and rigorously? 

Reviewer #1: No

Reviewer #2: Yes

Reviewer #3: Yes

3. Have the authors made all data underlying the findings in their manuscript fully available?

Reviewer #1: No

Reviewer #2: Yes

Reviewer #3: Yes

4. Is the manuscript presented in an intelligible fashion and written in standard English?

Reviewer #1: No

Reviewer #2: Yes

Reviewer #3: Yes

5. Review Comments to the Author

Reviewer #1: Dear author main theories that you have used in designing research framework are not technically linked and therefore unable to recommend your work , For instant TTF theory is not properly linked similarly, what you mean effort expectancy where ? in driving ? or adoption ?

Reviewer #2: The manuscript’s aims are sound, and the structure of the study, somewhat, is appropriate. Also, the findings of the study are considerable.

Nevertheless, I suggest the manuscript could be improved significantly by some revisions that are outlined below. I suggest the manuscript’s publications in your journal after the editors are satisfied the authors have made the suggested revisions. The suggested revisions (detailed below) include:

- In Introduction section, I suggest the authors clearly addressed that why he/she carried out this study? The gap should be addressed in introduction. However, the authors refer to the gap, but more information is needed.

- In the literature section, there are many study which addressed this issue. I suggest the authors mentioned them in literature review section.

- I suggest the manuscript could be improved significantly by more description of findings and cite to more document in each specific section.

- The contribution of knowledge is not clear, very well. The author should identify the lessons that we can learn from his/her study.

- The originally / novelty of the study is not clear very well.

Reviewer #3: 1)The introduction should provide a clearer explanation of the study's contributions.

2) While the literature review effectively discusses relevant theories, it's crucial to include a section detailing how the model was constructed based on these theories.

3) In the discussion section, clarify the reasons for either accepting or rejecting the hypotheses within the context, and support these decisions with reference to prior research. The discussion section requires further development.

4) Consider creating a dedicated section to address both the theoretical and practical implications.

5) Ensure a comprehensive conclusion that addresses the study's core questions.

6) For enhancing the UTAUT theory section, you can refer to the following sources:

Al-Mamary, Y. H. S. (2022). "Understanding the use of learning management systems by undergraduate university students using the UTAUT model: Credible evidence from Saudi Arabia." Published in the International Journal of Information Management Data Insights, 2(2), 100092.

Y. H. S., Siddiqui, M. A., Abdalraheem, S. G., Jazim, F., Abdulrab, M., Rashed, R. Q., ... & Aliyu Alhaji, A. (2023). "Factors impacting Saudi students’ intention to adopt learning management systems using the TPB and UTAUT integrated model." Published in the Journal of Science and Technology Policy Management.

6. PLOS authors have the option to publish the peer review history of their article (what does this mean?). If published, this will include your full peer review and any attached files.

Reviewer #1: **Yes: **Dr. Samar Rahi

Reviewer #2: No

Reviewer #3: No

---

## [Author Response · Author response to Decision Letter 0]

3 Jan 2024

Reviewer#1, Comment # 1: Dear author main theories that you have used in designing research framework are not technically linked and therefore unable to recommend your work, For instant TTF theory is not properly linked similarly, what you mean effort expectancy where ? in driving ? or adoption ?

Author response: Thanks for your concern. 

Author action: We appreciate your thoughtful review and constructive comments on the research presented. Your feedback has been instrumental in refining the theoretical foundations of the study. I would like to provide clarifications and address the concerns you raised regarding technical linkages of main theories, particularly TTF.

Integration of TTF with UTAUT theory

“This research integrates the UTAUT and TTF to explain the user adoption of EVs from both perspective including TTF and technology perceptions. According to Zhou et al. (2010) argued that individual variances of UTAUT (45.7%) and TTF (43.3%) lower than the integrated model of UTAUT-TTF (57.5%). This highlight the advantage of integrated model over individual models. Khashan et al. (2023) argued that the integration of both theories is crucial, emphasizing that user preference to adopt new technologies goes beyond mere perceptions. It also hinges on how effectively the characteristics of those technologies align with the tasks they are intended to assist users in completing. Several studies have found a strong association among TTF and UTAUT constructs, which may facilitate adoption in automotive (Kim et al., 2022), banking (Tarhini et al., 2016), construction (Cai et al., 2023), education (Tian and Yang, 2023), healthcare (Uymaz et al., 2023), and public sector (Alkhwaldi et al., 2022) as shown in Table 1. Nevertheless, there is a dearth of empirical evidence supporting the application of the integrated framework in the EVs domain. This points to a theoretical gap in previous research, which can be addressed by incorporating it into the current study to fill this void.” Furthermore, a the below table explain the integration of TTF and UTAUT in the prior literature. 

Theoretical framework Industry/sector type Country Sampling technique Summary of findings References

UTAUT and TTF Automotive South Korea None Both UTAUT-TTF can server an essential role in predicting consumer behaviour. Kim et al. (2022)

 Banking Lebanon and China Probability and Non-probability The outcomes highlight that both UTAUT and TTF are essential theoretical framework to explain the mobile banking user adoption. Tarhini et al. (2016) & Zhou et al. (2010)

 Construction China None The constructs of TTF have significant association with UTAUT variables. Cai et al. (2023) and Hilal et al. (2019)

 Education India, China, Malaysia Probability and Non-probability The integration of information system theories (UTAUT-TTF) promote students active learning and enable them to exchange information and knowledge more efficiently. Shirolkar and Kadam (2023), Tian and Yang (2023), Al-Rahmi et al. (2022), Al-Rahmi et al. (2022), Wang et al. (2022), Wan et al. (2020) and Chang (2013)

 Healthcare China, Turkey, Iran, South Korea, Pakistan Probability and Non-probability The findings indicated that UTAUT and TTF help to understand the user intention to adopt healthcare services. Uymaz et al. (2023), Abdekhoda et al. (2022), Kang et al. (2022), Gu et al. (2021), Wang et al. (2020)

 Retail USA Probability Both frameworks (UTAUT and TTF) positively explain the consumers intention to adopt technology-driven products. Park et al. (2015)

 Public sector Pakistan, Jordan, Yemen Probability and Non-probability Both UTAUT and TTF are the stronger predictors of BI than either UTAUT or TTF alone. Alkhwaldi et al. (2022), Shahbaz et al. (2021) and Isaac et al. (2019)

 Services Tiawan None Both theoretical frameworks are important to adopt new and innovative technologies. Pai and Tu (2011)

The section of theoretical framework has completely revise with updated references from Pg-4-6, lines- 108-186 and for Table-1 on Pg- 20-21.

Clarification on constructs (effort expectancy)

I understand your concern about the ambiguity surrounding the term "effort expectancy." To provide clarity, the term is specifically used in the context of adoption, reflecting users' perceived ease of use and the effort required to adopt the technology under investigation. This clarification has been added to the relevant sections of the manuscript to avoid any potential misunderstanding. Additionally, the following table explain the constructs in the theoretical model.

Constructs Definition Construct sources Dimension involved Factors related to the constructs Reference

Performance expectancy PE refers to the degree to which individual believe that utilizing the technology will empower them to achieve optimal improvement in job performance. UTAUT Technology Perceived usefulness, and view on performance Shin et al., (2022), Abdelmageed & Zayad (2020), and Cai et al., (2023)

Effort expectancy EE pertains to the ease of effort associated with the utilization of technology. UTAUT Technology Perceived ease of use, technical complexity, and additional efforts Cai et al., (2023), Jang et al., (2022)

Facilitating condition FC involve the extent to which individuals are confident that the current organizational and technical infrastructure is in place to support the utilization of the technology. UTAUT Technology/organization Experience and skills, R&D, and organizational structure Zakaria et al., (2018), Luo et al., 2019)

Social influence The belief that individual actions related to the use of technology will be accepted and embraced by affluent individuals or peers is referred to as social influence. UTAUT Environment Government policy, stakeholder norm, and industry chain Wang et al. (2021), Mao et al. (2018)

Task Characteristics Task characteristics encompasses the challenges that potential adopters need to manage when employing technology in a specific context. TTF Technology/Environment Cost, standardization, and coordination Wu et al. (2019), and Masood et al. (2021)

Technology Characteristics Technology characteristics encompasses the benefits offered by the technology. TTF Technology Relative advantage, and motivation Hu et al. (2019), and Cai et al. (2023)

Task-Technology fit The extent to which the technology is perceived to complete the particular tasks. TTF Technology Compatibility, and suitability Gao et al. (2018)

Adoption intention The intention of (potential) adopters for technology adoption. UTAUT ------- Willingness Cai et al. (2023), and Khashan et al. (2023)

We believe that these revisions will strengthen the overall theoretical framework and improve the clarity of the connections between different theories and concepts. I appreciate your diligence in reviewing my work and thank you for guiding me towards enhancing the quality of the research.

Reviewer#2, Comment # 1: In Introduction section, I suggest the authors clearly addressed that why he/she carried out this study? The gap should be addressed in introduction. However, the authors refer to the gap, but more information is needed.

Author response: Thanks for highlighting the issues and concerns. 

Author action: Thank you for your insightful feedback. We appreciate your suggestion regarding the need for a more explicit articulation of the study's rationale and the addressed gap in the introduction. In response, we have revised the introduction to provide a clearer and more detailed explanation of why this study was undertaken and how it fills the identified gap in the existing literature. We trust that these modifications enhance the overall coherence and purpose of the manuscript.

According to Miles (2017), the research work has multiple gaps like literature, population, theoretical, evidence, practical-knowledge and methodological gaps. Miles (2017) further argued that researchers should highlight three to four gaps. Thus, this research will fill the literature, theoretical, population and evidence gaps. 

Literature and evidence gap: “Conclusively, the global rise of EV adoption reveals a notable gap, especially in Middle East region, due to geopolitical and pandemic disruptions, emphasizing the urgency for targeted interventions to facilitate a more inclusive and sustainable EV transition.” The statement has been added on Pg-3, line 71-73.

Population gap: “In a nutshell, despite the pressing need for Saudi Arabia to reduce CO emissions, there exists a significant gap in the country transition to EV, exacerbated by a historical dependence on fossil fuels, limited use of renewable energy in road transportation, and the challenges of aligning consumer incentives with the complex dynamics of technological transformation and consumer acceptance.” The statement has been added on Pg-3, line 88-92.

Theoretical gap: “Amidst evolving EV adoption literature, there is a growing recognition for diverse framework beyond traditional theories, necessitating deeper exploration of individual and technology dynamics.” The statement has been added on Pg-4, line 105-107. Furthermore, in the theoretical framework section a conclusive statement has been added on Pg-6, lines-192-194, “Table 1. Nevertheless, there is a dearth of empirical evidence supporting the application of the integrated framework in the EVs domain. This points to a theoretical gap in previous research, which can be addressed by incorporating it into the current study to fill this void.”

Reviewer#2, Comment # 2: In the literature section, there are many study which addressed this issue. I suggest the authors mentioned them in literature review section.

Author response: Thanks for highlighting the issues and concerns. 

Author action: We have addressed your suggestion by incorporating a comprehensive literature review section within the theoretical framework. To provide a more detailed overview, we have also included supplementary material that features a literature review table, the same has also been added at the end of this document which offering a thorough examination of studies addressing the pertinent issue. We believe these additions strengthen the scholarly foundation of the manuscript. 

Reviewer#2, Comment # 3: I suggest the manuscript could be improved significantly by more description of findings and cite to more document in each specific section.

Author response: Thanks for your concern. 

Author action: We have revisited the manuscript and made substantial improvements by providing more detailed descriptions of our findings in each specific section. Additionally, we have expanded our citations to include a more comprehensive range of relevant documents, thereby enhancing the overall depth and robustness of our work. For instance, on Pg-13 & 14, lines 423-429, 442-445 and 447-450.

Reviewer#2, Comment # 4: The contribution of knowledge is not clear, very well. The author should identify the lessons that we can learn from his/her study.

Author response: Thanks for your suggestions.

Author action: Thank you for your thoughtful feedback. We acknowledge the importance of clarifying the knowledge contribution of our study. We have revised the theoretical contribution section on Pg-15, line 462-478 to explicitly identify the lessons derived from our research.

The study makes significant contributions by broadening the understanding of electric vehicles (EVs) through the integration of TTF and UTAUT frameworks. This synthesis offers a fresh perspective beyond a single theoretical framework, addressing a recognized call in literature. Through a methodical empirical approach, the study enhances the field of technological acceptance models, providing a validated model with robust associations.

Furthermore, our research contributes to the literature by examining specific challenges and opportunities in the Middle East region, particularly in Saudi Arabia, related to EV adoption. The study provides nuanced insights and proposes targeted interventions to facilitate a more inclusive and sustainable transition to EVs in this unique and dynamic environment.

Finally, empirical research makes a significant contribution by addressing a theoretical gap in the EV adoption literature. The application of an integrated framework of UTAUT and TTF fills an existing void in research, enriching and validating the theoretical landscape. These contributions provide valuable insights for future research and practical applications in the EV domain.

We believe these clarifications emphasize the lessons and knowledge gained from our study. Thank you for your time and valuable feedback.

Reviewer#2, Comment # 5: The originally / novelty of the study is not clear very well.

Author response: Thanks for your concerns.

Author action: Thank you for your feedback. We appreciate the opportunity to clarify the novelty and originality of our study. To address this concern, we have revised both the theoretical and practical implication sections of the manuscript on Pg-15 & 16, lines-462 to 494.

In the theoretical section, we have explicitly highlighted the identified gaps stated in the introduction and theoretical framework. The integration of TTF and UTAUT frameworks to offer a fresh perspective addresses the existing call in the literature and contributes to the theoretical landscape. Our revised theoretical framework emphasizes the unique approach taken to fill the identified gaps.

Furthermore, in the practical implications section, we have underscored how the study's findings provide targeted interventions to address specific challenges and opportunities in the Middle East region related to EV adoption, aligning with the identified gaps in the introduction and theoretical sections. This alignment emphasizes the practical relevance and contributions of our research.

We trust that these revisions clarify the novelty and originality of our study by explicitly linking the identified gaps with our theoretical and practical contributions. Thank you for your time and valuable feedback.

Reviewer#3, Comment # 1: The introduction should provide a clearer explanation of the study's contributions

Author response: Thanks for your concerns.

Author action: Thank you for your feedback. We have carefully revised the introduction to provide a clearer and more detailed explanation of the study's contributions. The updated introduction explicitly outlines the unique contributions of our research, emphasizing the identified gaps in the literature and how our study fills these gaps by integrating TTF and UTAUT frameworks. We believe these revisions enhance the overall clarity of the study's contributions. The contributions statement has been added on Pg-4, line-116 to 124.

Reviewer#3, Comment # 2: While the literature review effectively discusses relevant theories, it's crucial to include a section detailing how the model was constructed based on these theories.

Author response: Thanks for your concerns.

Author action: 

Reviewer#3, Comment # 3: In the discussion section, clarify the reasons for either accepting or rejecting the hypotheses within the context, and support these decisions with reference to prior research. The discussion section requires further development.

Author response: Thanks for your concerns.

Author action: We have addressed your suggestion by incorporating a comprehensive literature review section within the theoretical framework. To provide a more detailed overview, we have also included supplementary material that features a literature review table, the same has also been added at the end of this document which offering a thorough examination of studies addressing the pertinent issue. We believe these additions strengthen the scholarly foundation of the manuscript.

Reviewer#3, Comment # 4: Consider creating a dedicated section to address both the theoretical and practical implications.

Author response: Thanks for your concerns.

Author action: Thank you for your insightful suggestion. We have carefully considered your feedback and have created a dedicated section in the manuscript to address both the theoretical and practical implications of our study on Pg-15 & 16, lines 462-494. This new section aims to provide a more structured and comprehensive overview of the contributions our research makes to both theoretical advancements and practical applications.

Reviewer#3, Comment # 5: Ensure a comprehensive conclusion that addresses the study's core questions.

Author response: Thanks for your concerns.

Author action: We have included the conclusion section that addresses the core questions on Pg-16, line 514-520. 

Reviewer#3, Comment # 6: For enhancing the UTAUT theory section, you can refer to the following sources:

• Al-Mamary, Y. H. S. (2022). "Understanding the use of learning management systems by undergraduate university students using the UTAUT model: Credible evidence from Saudi Arabia." Published in the International Journal of Information Management Data Insights, 2(2), 100092.

• Y. H. S., Siddiqui, M. A., Abdalraheem, S. G., Jazim, F., Abdulrab, M., Rashed, R. Q., ... & Aliyu Alhaji, A. (2023). "Factors impacting Saudi students’ intention to adopt learning management systems using the TPB and UTAUT integrated model." Published in the Journal of Science and Technology Policy Management.

Author response: Thanks for your concerns.

Author action: The valuable research documents have been added in the revised manuscript on Pg-17, line 541-546.

---

## [Editor Report · Decision Letter 1]

15 Jan 2024

From Theory to Practice: An integrated TTF-UTAUT study on Electric Vehicle Adoption Behavior

PONE-D-23-27266R1

Dear Dr. Ali,

We’re pleased to inform you that your manuscript has been judged scientifically suitable for publication and will be formally accepted for publication once it meets all outstanding technical requirements.

Kind regards,

Sudarsan Jayasingh, Ph.D

Academic Editor

PLOS ONE
---

## [Editor Report · Acceptance letter]

29 Feb 2024

PONE-D-23-27266R1 

PLOS ONE

Dear Dr. Ali, 

I'm pleased to inform you that your manuscript has been deemed suitable for publication in PLOS ONE. Congratulations! Your manuscript is now being handed over to our production team.

Kind regards, 

on behalf of

Dr. Sudarsan Jayasingh 

Academic Editor

PLOS ONE